# Lambda-PFLOTRAN 1.0: Workflow for Incorporating Organic Matter Chemistry Informed by Ultra High Resolution Mass Spectrometry into Biogeochemical Modeling

Katherine A. Muller[1], Peishi Jiang[1], Glenn Hammond[1], Tasneem Ahmadullah[1], Hyun-Seob Song[2], Ravi Kukkadapu[1], Nicholas Ward[3], Madison Bowe[3], Rosalie K. Chu[1], Qian Zhao[1], Vanessa A. Garayburu-Caruso[1], Alan Roebuck[3], Xingyuan Chen[1]

[1] Pacific Northwest National Laboratory, Richland, WA 99352, USA
[2] Department of Biological Systems Engineering, University of Nebraska—Lincoln, Lincoln, Nebraska, USA
[3] Pacific Northwest National Laboratory, Sequim WA 98382, USA

*Correspondence to*: Katherine Muller (katherine.muller@pnnl.gov)

For submission to Geoscientific Model Development

**Abstract.** Organic matter (OM) composition plays a central role in microbial respiration of dissolved organic matter and subsequent biogeochemical reactions. Here, a direct connection of organic matter chemistry and thermodynamics to reactive transport simulators has been achieved through the newly developed Lambda-PFLOTRAN workflow tool that succinctly incorporates carbon chemistry data generated from Fourier transform ion cyclotron resonance mass spectrometry (FTICR-MS) into reaction networks to simulate organic matter degradation and the resulting biogeochemistry. Lambda-PFLOTRAN is a python-based workflow, executed through a Jupyter Notebook interface, that digests raw FTICR-MS data, develops a representative reaction network based on substrate-explicit thermodynamic modeling (also termed lambda modeling due to its key thermodynamic parameter λ used therein), and completes a biogeochemical simulation with the open source, reactive flow and transport code PFLOTRAN. The workflow consists of the following five steps: configuration, thermodynamic (lambda) analysis, sensitivity analysis, parameter estimation, and simulation output and visualization. Two test cases are provided to demonstrate the functionality of the Lambda-PFLOTRAN workflow. The first test case uses laboratory incubation data of temporal oxygen depletion to fit lambda parameters (i.e., maximum utilization rate and microbial carrying capacity). A slightly more complex second test case fits multiple lambda formulation and soil organic matter release parameters to temporal greenhouse gas generation measured during a soil incubation. Overall, the Lambda-PFLOTRAN workflow facilitates upscaling by using molecular-scale characterization to inform biogeochemical processes occurring at larger scales.

## 1 Introduction

Microbial respiration of dissolved organic carbon (DOC) is a main driver of environmental biogeochemical processes. Mechanistic biogeochemical models often rely on lumping organic matter into a few distinct carbon pools (e.g., dissolved, sorbed, mineral associated or refractory, labile, etc.) (e.g., Fatichi, et al., 2019, Robertson et al., 2019, Wang et al., 2013) but do not fully consider the properties of the organic matter (OM) compounds individually. Pooled carbon approaches have benefits, such as assigning variable levels of bioavailability, however, this approach does not capture the complex temporal dynamics of respiration driven by OM composition, as aerobic respiration rates have been linked to organic carbon concentration, thermodynamics of the OM (Stegen et al., 2018, Garayburu-Caruso et al., 2020), as well as the diversity of OM compounds present (Lehmann et al. 2020, Stegen et al., 2022). Such findings highlight the importance of incorporating individual OM chemistry into biogeochemical modeling to capture, and ultimately predict, system behavior more accurately.

There are many advanced instrumentation techniques capable of detecting and identifying individual OM formulae that comprise a bulk OM sample (e.g., GC-MS, HPLC-MS, Fourier transform ion cyclotron resonance mass spectrometry [FTICR-MS], etc.). For instance, FTICR-MS is a powerful, high-resolution, method that identifies molecular formulae for individual organic compounds. In any given environmental sample, FTICR-MS (or other ultra high-resolution methods) will typically resolve thousands of discrete OM molecular formulae, each with a unique mass and elemental composition (Cooper et al., 2020, Bahureksa et al., 2021). However, untargeted analytical techniques like FTICR-MS are only able to determine if a compound is present and cannot quantify the total concentration associated with each organic matter molecule. Still, such techniques do provide immense amounts of characterization data encompassing a deeper analytical window than measuring a small number of individual biomarkers quantitatively (e.g., Ward et al., 2013). Utilizing such high-resolution molecular data in reactive transport modeling frameworks affords new opportunity to advance carbon cycling in terrestrial, riverine and coastal systems despite of various theoretical and computational challenges.

Substrate-explicit thermodynamic modeling (SXTM) provides an avenue for incorporating individual OM reactivity based on thermodynamics (Song et al., 2020) into reactive transport models. The SXTM procedure takes the individual chemical formula derived from FTICR-MS (or another high-resolution technique) and uses its thermodynamic properties to generate an oxidation reaction for each molecular formula present in a sample. The corresponding reaction stoichiometry is then determined by considering catabolic, anabolic, and metabolic reactions and balancing energy for the overall metabolic reaction, allowing for the development of an aerobic respiration expression for each OM formula.

Still, the sheer number of compounds identified in each sample proves difficult for model integration. Typically, reactive transport simulators consider only a small number of primary species in their reaction networks, and most could not support modeling each of the thousands of organic matter molecules individually. Here, the developed Lambda-PFLOTRAN workflow addresses this challenge through grouping, or binning, similar compounds based on

their thermodynamic properties, allowing for the number of species considered within the reaction network to be
reduced, and thus decreasing the required computational resources.
Lambda-PFLOTRAN is a python-based workflow that digests raw FTICR-MS data, develops a representative reaction
network based on substrate-explicit thermodynamic modeling (Song et al., 2020), and completes a biogeochemical
simulation with the open source, parallel reactive flow and transport code, PFLOTRAN (Hammond et al., 2014).
PFLOTRAN is developed under an open source, GNU LGPL license. The term 'lambda' is used here because $\lambda$ is a
key parameter in the SXTM, which quantifies thermodynamic favorability of aerobic respiration of OM. The
connection between the unique reaction network developed for each FTICR-MS sample hinges on the use of
PFLOTRAN's reaction sandbox capability (Hammond, 2022). The reaction sandbox gives the ability to define
additional custom, kinetic reactions beyond standard formulations (e.g., mineral precipitation-dissolution, Michaelis-
Menten, etc.). The Lambda-PFLOTRAN workflow enables upscaling by using molecular-scale information to inform
larger scale biogeochemical processes occurring throughout a watershed which can be simulated with PFLOTRAN.
Herein we describe the Lambda-PFLOTRAN workflow process including the governing expressions, workflow steps,
data requirements, as well as the associated assumptions and limitations. Two illustrative test cases are also included
to demonstrate the workflow.
**2 Methods**
**2.1 Conceptual Model**
Respiration modeling herein is based on thermodynamic theory by Desmond-Le Quemener and Bouchez (2014) which
was updated for multiple OM formulas by Song et al. (2020). The generalized form of OM molecule is assumed to
take the form of $C_aH_bN_cO_dP_eS^{z}_f$. Each molecular formula then undergoes respiration (i.e., reaction with oxygen) based
on the following general reaction expression:
$y_{OM_i}OM_i + y_{H_2O}H_2O + y_{HCO_3^-}HCO_3^- + y_{NH_4^+}NH_4^+ + y_{HPO_4^{--}}HPO_4^{--} + y_{HS^-}HS^- + y_{H^+}H^+ + y_{e^-}e^- +$
$y_{O_2}O_2 + y_BBM = 0,$                            *(1)*
This generalized expression is used to describe the oxidation of any OM molecule, *i,* and has been normalized to one
mole of biomass (BM) produced. BM is assumed to have a formula of $CH_{1.8}O_{0.5}N_{0.2}$ (Stephanopoulos et al.,
1998; Kleerebezem and Van Loosdrecht, 2010). $OM_i$ represents the OM molecules as informed by FTICR-MS. Each
*y* represents the reaction stoichiometry for that reactant (*y* < 0) or product (*y* > 0). While this expression is specific for
cases where oxygen is the electron acceptor, such an expression could be updated for alternative electron acceptors.
Substrate-explicit thermodynamic modeling expressions developed from Song et al. (2020) were implemented in a
reaction sandbox within PFLOTRAN. The expressions were implemented in a general manner allowing for flexibility
in handling variations in FTICR-MS data and several user adjustable analysis configurations.
The microbial growth kinetics are described by Eq. (2):
$\mu_i{}^{kin} = \mu^{max} exp(-\frac{\alpha|y_{OM,i}|}{1000V_h[OM_{,i}]})exp(-\frac{\alpha|y_{O2,i}|}{1000V_h[O_2]})$,               (2)
where $\mu_i{}^{kin}$ is the unregulated uptake rate of reaction for $OM_i$ [hr$^{-1}$], $\mu^{max}$ is the maximal microbial growth rate [hr$^{-1}$
], $y_{OM,i}$ is the stoichiometry for $OM_i$ [mol-OM · mol-biomass$^{-1}$], $V_h$ is microbial harvest volume [m$^3$]. Given the
physical interpretation of $V_h$ as the microbial harvest volume, it is assumed here that the value of $V_h$ is the same for
both $OM_i$ and $O_2$, $[OM_i]$ is the organic matter concentration of $OM_i$ [mol-OM·L$^{-1}$], $y_{O_2,i}$ is the stoichiometry for $O_2$
for respiration of $OM_i$ [mol-O$_2$·mol-biomass$^{-1}$], $[O_2]$ is oxygen concentration [mol-O$_2$·L$^{-1}$], α is a microbial unit
conversion [mol-biomass] and 1000 is the conversion of m$^3$ to L.
Further, using a cybernetic modeling approach (after Song et al., 2018), all the unregulated uptake rates ($\mu_i{}^{kin}$) are
normalized by the sum of unregulated uptake rates across all reactions, *i* following Eq. (3):
$u_i = \frac{\mu_i{}^{kin}}{\sum_{i=1}^{n} \mu_i{}^{kin}}$                    (3)
where $u_i$ is the fraction of the unregulated rate [-]. The final regulated rate, $r_i$ [hr$^{-1}$] for each reaction is then computed
following Eq. (4):
$r_i = u_i\mu_i{}^{kin}$,                        (4)
For implementation within PFLOTRAN, the use of inhibition terms was required to prevent negative concentrations
once a reactant is nearly depleted.  For a reaction to proceed, all reactant species must be present above a minimum
concentration even if the molecules do not explicitly control the respiration rate (i.e., species other than OM and $O_2$,
Eq. (2). If a reactant concentration falls below a threshold concentration, the respiration rate is inhibited. Reactant
inhibition is computed by Eq. 5 (Kinzelbach et al., 1991) for reactant species *j:*
$I_j = 0.5 + \frac{arctan([C_j]-C_{thj})\cdot f}{\pi}$,             (5)
where $C_{th,i}$ is the threshold concentration [M], *f* is the threshold scaling factor [-]. The default $C_{th_j}$ is 10$^{-20}$ M.
The reaction rates are also inhibited by the microbial carrying capacity of the system, $I_{cc}$, as follows in Eq. (6):
$I_{CC} = 1 - \frac{[BM]}{CC}$                  (6)
where [BM] is the biomass concentration [mol-BM·L$^{-1}$], CC is the biomass carrying capacity [mol-BM·L$^{-1}$]. $I_{cc}$ has a
non-negativity constraint, so if [BM] > CC, then $I_{cc} = 0$.
These inhibition factors are applied to the overall rate expression as shown in Eq. (7).
$r_{i,inhibited} = r_i I_{CC} \prod I_j \quad \forall\, y_{i,j} < 0,$                               (7)
The overall individual species rates, d[$C_j$]/dt, [mol-species·$L^{-1}$·$hr^{-1}$] are then computed as follows with Eq. (8):
$\frac{dC_j}{dt} = \left(\sum_{i=1}^{n} y_{i,j} r_{i,inhibited}\right)[BM],$                                  (8)
where $j$ is the species index. The total number of species includes 7 general species (i.e., $HCO_3^-$, $NH_4^+$, $HPO_4^-$, $HS^-$,
$H^+$, $O_2$, BM (i.e., Eq (1)) and the OM species considered (i.e., typically 10). $i$ is the reaction index, $n$ is total number
of reactions as based on the total number of OM species (typically, with this workflow $n$ =10). $y_{i,j}$ is the coefficient
for species $j$ in reaction $i$.
The expression for biomass is also modified to account for biomass decay (note all biomass stoichiometries are 1 by
definition):
$\frac{dBM}{dt} = \left(\sum_{i=1}^{n} y_{i,j} r_{i,inhibited}\right)[BM] - k_{deg}[BM],$                      (9)
where $k_{deg}$ is the biomass decay rate [$hr^{-1}$].

**2.2 Lambda Analysis and Binning**
To reduce the number of organic compounds considered in the simulation, OM molecules are grouped, or binned,
based on their λ value computed by Eq. (10):
$\lambda = \frac{\Delta G_{r,anabolic} + \Delta G_{r,dissipation}}{(-\Delta G_{r,catabolic})},$                                 (10)
where $\Delta G$ are the Gibbs energies for the anabolic and catabolic reactions and the associated dissipation energy,
respectively. The value of λ is indicative of how many times the catabolic reaction needs to be completed to provide
the energy required to synthesis one mole of biomass. Lower λ values suggest higher thermodynamic favorability of
OM respiration. Using the chemical formula determined for each OM molecule, the energy balance equations are
solved providing the overall reaction stoichiometry Eq. (1) and the λ is calculated. Using the λ value for each molecule,
the cumulative probability distribution for the sample is produced (Figure 2).

**Figure 1:** Flow Chart of the Lambda-PFLOTRAN Workflow.

It is this conversion from individual compounds to a distribution that is critical for reducing the entire sample down to a representative set of expressions. The λ bins are then formed by splitting the cumulative probability distribution into equally weighted sections as which to define the overall sample by. The illustrative example shown in Fig. 2 demonstrates the sample distribution being divided into 10 sections (i.e., in this case each section contains 10% of the overall sample distribution).

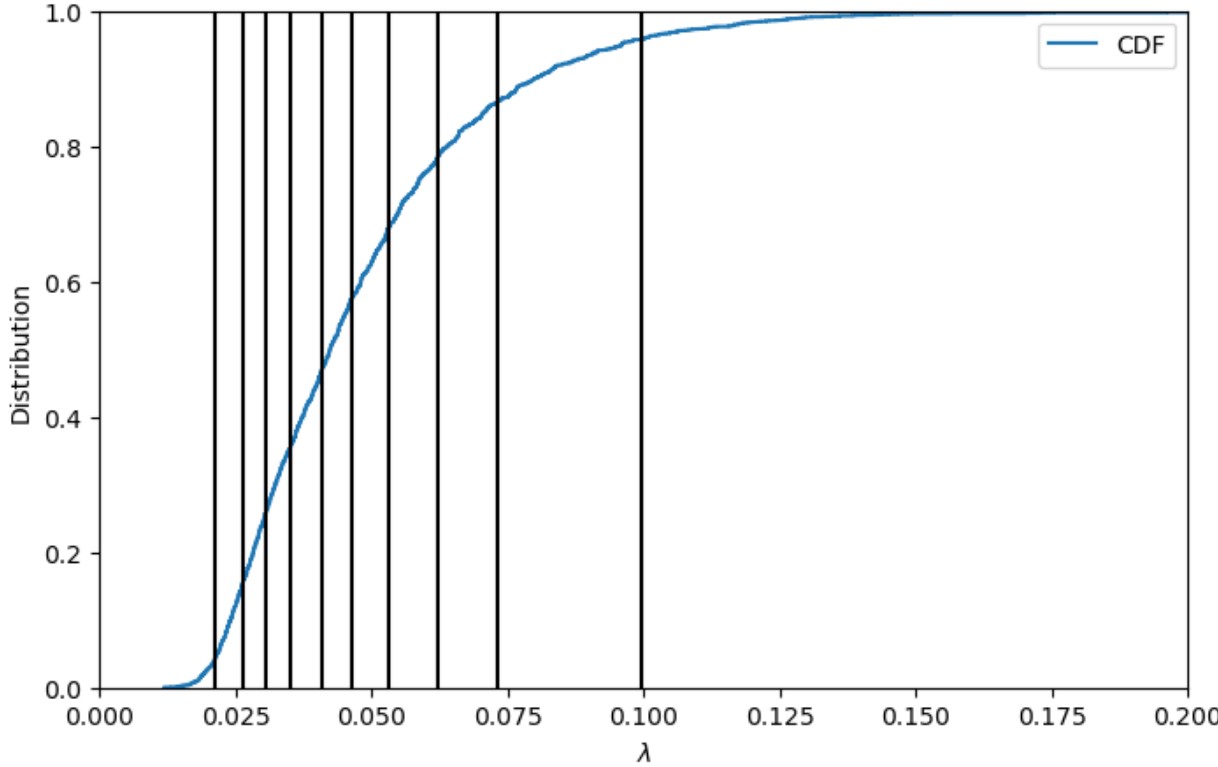

152

**Figure 2:** Lambda binning to convert raw FTICR-MS into a representative reaction network using the cumulative probability distribution function (CDF) for Test Case 1a. Vertical lines display the average λ value for each of the 10 bins (left to right, λ bin 1 to 10).

Each section is used to determine a representative organic matter formula and the associated reaction and stoichiometry of that λ bin. The group of representative reactions (one per bin) is called the reaction network. A demonstrative reaction network defined by λ analysis and binning is shown in Table 1.

**Table 1:** Reaction Network Developed from Lambda Theory for Test Case 1a

| Bin Number | Representative Organic Matter Species Formula | λ | $y_{OM}$ | $y_{HCO3^-}$ | $y_{NH4^+}$ | $y_{HPO4^{--}}$ | $y_{HS^-}$ | $y_{H^+}$ | $y_{O2}$ |
|---|---|---|---|---|---|---|---|---|---|
| 1 | $C_{31}H_{44}N_{0.33}O_{4.8}P_{0.6}S_{0.3}$ | 0.021 | -0.05 | 0.64 | -0.17 | -0.18 | 0.03 | 0.02 | -1.07 |
| 2 | $C_{26}H_{39}N_{0.20}O_{7.0}P_{0.6}S_{0.1}$ | 0.026 | -0.07 | 0.68 | -0.10 | -0.19 | 0.04 | 0.01 | -1.06 |
| 3 | $C_{22}H_{36}N_{0.24}O_{7.5}P_{0.5}S_{0.1}$ | 0.031 | -0.08 | 0.69 | -0.02 | -0.18 | 0.04 | 0.01 | -1.06 |
| 4 | $C_{20}H_{32}N_{0.28}O_{7.3}P_{0.4}S_{0.1}$ | 0.035 | -0.08 | 0.72 | -0.08 | -0.18 | 0.04 | 0.01 | -1.05 |
| 5 | $C_{19}H_{29}N_{0.48}O_{7.9}P_{0.3}S_{0.2}$ | 0.041 | -0.09 | 0.79 | -0.17 | -0.16 | 0.03 | 0.02 | -1.04 |
| 6 | $C_{18}H_{26}N_{0.68}O_{8.1}P_{0.2}S_{0.2}$ | 0.046 | -0.10 | 0.85 | -0.27 | -0.13 | 0.02 | 0.02 | -1.03 |
| 7 | $C_{17}H_{24}N_{0.69}O_{8.1}P_{0.2}S_{0.2}$ | 0.053 | -0.11 | 0.90 | -0.32 | -0.12 | 0.02 | 0.02 | -1.02 |
| 8 | $C_{15}H_{20}N_{0.67}O_{7.6}P_{0.2}S_{0.2}$ | 0.062 | -0.13 | 0.94 | -0.42 | -0.11 | 0.02 | 0.03 | -1.00 |

| 9 | $C_{13}H_{19}N_{1.13}O_{87.4}P_{0.1}S_{0.2}$ | 0.073 | -0.15 | 1.01 | -0.48 | -0.03 | 0.01 | 0.03 | -1.00 |
| 10 | $C_{10}H_{15}N_{1.56}O_{6.5}P_{0.1}S_{0.2}$ | 0.100 | -0.21 | 1.17 | -0.75 | 0.12 | 0.01 | 0.04 | -0.97 |


Currently, the representative OM molecule that defines each bin is computed as the average chemical formula of all
the molecules present in that λ section. The disadvantage of this approach is that unrealistic compounds are defined
as representative molecules instead of realistic molecules. The issue with selecting a single, but real compound, from
within each λ section resides in chemical complexity and variation - for instance some molecules may contain low
levels of phosphorous or sulfur and others may not contain either element in the chemical formula. Thus, requiring
the representative chemical formula to be a real compound present in the sample would create basis which would
propagate through the reaction network and into the resulting biogeochemical simulation results.
**2.3 Lambda-PFLOTRAN Workflow**
The Lambda-PFLOTRAN workflow digests raw FTICR-MS data, calculates the λ distribution for the sample,
generates the λ bins and corresponding reaction network, and completes a biogeochemical simulation using
PFLOTRAN. Further, we incorporated sensitivity analysis and ensemble data assimilation to enable an in-depth
exploration of the impact of reaction parameters on respiration as well as a straightforward parameter estimation
method to fit model parameters to experimental data.
The workflow is implemented through a user-friendly Jupyter notebook interface (Kluyver et al., 2016) where a user
can configure the simulation parameters by adjusting initial concentrations, λ binning configuration, parameter values
and/or ranges, and data assimilation options. Based on the user's data file and the associated parameters, scripts within
the Jupyter notebook write the corresponding PFLOTRAN input files, including OM molecules and aqueous
chemistry. The PFLOTRAN simulations are completed locally through a Docker container making this capability
much more user-friendly and accessible. The progress of the data assimilation tool used for parameter fitting is
illustrated within the Jupyter notebook. The resulting best fit final biogeochemical simulation is output visually with
plots and as a text file (when applicable).
The Lambda-PFLOTRAN workflow steps are shown in Figure 1 and described in detail in the following subsections:
**2.3.1 Step 1 – Workflow configuration**
The first step is to set up the workflow configuration for a Lambda-PFLTORAN application. This includes specifying
the file and folder locations of the following information: 1) FTICR-MS raw data file (.csv), 2) initial species
concentrations file (.csv) that includes starting molar concentrations for $HCO_3^-$, $NH_4^+$, $HPO_4^{2-}$, $HS^-$, $H^+$, $O_2$ (aq), BM
and total organic carbon (TOC), 3) PFLOTRAN database template file, 4) PFLOTRAN executable file, 5) workflow
output folder, and if completing parameter estimation, (6) the data observation file (.csv), if applicable.
The user is also asked to configure workflow settings related to: (1) the lambda analysis configuration, including
number of λ bins and method to define the λ bins (i.e., cumulative vs uniform); (2) the respiration modeling parameter
setup, including the list of the parameters to be estimated and their associated upper and lower bounds and (3) the data
assimilation configuration (see below).
**2.3.2 Step 2 – Organic Matter Chemistry using Lambda Analysis**
With only an input of FTICR-MS data, the workflow first performs the lambda analysis (Section 2.2) to group OM
molecules into various λ bins based on each compound's thermodynamics (Figure 2) and produce the corresponding
reaction network for respiration (Table 1). The default number of λ bins is 10, although this can be adjusted in the
workflow configuration by the user, if desired. The generated reaction network is then automatically parsed by the
workflow into a text file that can be read by PFLOTRAN.
**2.3.3 Step 3 – Sensitivity Analysis using Mutual Information**
This step performs the global sensitivity analysis on the parameters to be estimated. Ensemble parameters are first
generated by randomly sampling from their predefined ranges in the configuration step and saved into an HDF5 file.
Then, the workflow generates a PFLOTRAN input deck to conduct ensemble simulations using the ensemble
parameters. The generated ensemble model states enables a global sensitivity analysis using mutual information
(Cover and Thomas, 2006; Jiang et al, 2022) as follows:
$I(X;Y) = H(Y) - H(Y|X) = \sum_{X=x} \sum_{Y=y} p(x,y) log\left(\frac{p(x,y)}{p(x)p(y)}\right),$     (11)
where $x$ and $y$ are the specific values of $X$ and $Y$, respectively; $H(Y)$ is the Shannon's entropy of $Y$; $H(Y|X)$ is the
conditional entropy of $Y$ given $X$; $p$ is the probability density function. Higher $I$ indicate stronger sensitivity between
$X$ and $Y$. Besides sensitivity analysis, the ensemble parameter/states also serve as the prior information for parameter
estimation at the next step.
**2.3.4 Step 4 – Parameter Estimation using Ensemble Smoother for Multiple Data Assimilation**
The workflow adopts Ensemble Smoother for Multiple Data Assimilation (Emerick and Reynolds, 2013; Jiang et al,
2021), abbreviated as ESMDA, for data assimilation in this step. Rooted in ensemble Kalman filter, ESMDA is an
iterative data assimilation approach that assimilates the observations on the entire time period for multiple times to
reduce the uncertainty of the estimated or posterior parameters. During each iteration of ESMDA, the model
parameters are updated based on the following equation:
$m_{k,l}^{u} = m_{k,l}^{f} + C_{MD,l}^{f}\left(C_{DD,l}^{f} + \alpha_l C_D\right)^{-1}\left(d_{obs} + \sqrt{\alpha_l}C_D^{\frac{1}{2}}z_k - d_{k,l}^{f}\right), \; k = 1, ..., N_e \; and \; l = 1, ..., L,$     (12)
where the subscripts $k$ and $l$ are the indices of the ensemble member and the iteration, respectively; the superscripts $u$
and $f$ are the updated and forecast parameters or states, respectively; $N_e$ is the number of ensemble members; $L$ is the
number of iterations; $m_{k,l}^{f}$ and $m_{k,l}^{u}$ are the kth ensemble member of the forecast/prior and updated/posterior
parameters, respectively, at the $l$th iteration; $d_{obs}$ is the observation; $z_k$ is the observation noise sampled from
independent standard normal distributions for the $k$th ensemble member; $d^f_{k,l}$ is the $k$th ensemble member of the
predicted observation states by the model using $m^f_{k,l}$; $C^f_{MD,l}$ is the cross-covariance matrix between the prior parameters
$m^f_l$ and the predicted observation states $d^f_l$; $C^f_{DD,l}$ is the auto-covariance matrix of the predicted observation states $d^f_l$;
$C_D$ is the auto-covariance matrix of the observation error; and $\alpha_l$ is the inflation coefficient at the $l$th iteration with the
sum of all $\alpha_l$ equal to one.

Here, the assimilation starts with taking the ensemble model parameters/states in Step 3 and the provided observations,
and calculates the posterior parameters using ensemble Kalman filter, updates the prior parameters with the current
posterior for the next iteration, and then repeats the whole process for multiple times (typically 3 to 5 iterations, as
defined by the user). The final estimated parameters are obtained from the posterior parameter at the last iteration and
are updated in the parameter HDF5 file. The parameter estimation is implemented in a way that allows assimilating
either a single (e.g., Test Case 1) or multiple observed species simultaneously through a simple change of the inputs.
For example, if temporal experimental or field data is available for oxygen, pH, and total carbon, all these data sources
could be simultaneously fit to, with only minor adjustments to Jupyter notebook.
**2.3.5 Step 5 – Simulation Output and Visualization**
The last step performs the ensemble simulation of the biogeochemical modeling a final time using the estimated
parameters in Step 4. Optionally, users can further pick the realization with the best performance. The user has the
option to select their preferred goodness of fit metric from the following options as a means for selecting the best
performing simulation: R-squared ($R^2$), Root Mean Squared Error (RMSE), Modified Kling-Gupta Efficiency
(mKGE), Nash-Sutcliffe Model Efficiently Coefficient (NSE), or Correlation Coefficient (CorC). Based on the
selection, the final time series of aqueous chemistry, oxygen consumption, $CO_2$ production, and lambda binned, and
total organic carbon concentrations will be computed and plotted.
**3 Test Cases**
**3.1 Test Case 1 - Oxygen Depletion Incubation Experiments.**
In the first illustrative example, the workflow was used to fit $\mu_{max}$ to laboratory incubation experiments where oxygen
levels were measured over two hours in a closed reactor. The incubation experiments were completed as part of the
Worldwide Hydrobiogeochemistry Observation Network for Dynamic River Systems (WHONDRS) program
(Goldman et al, 2020). For these incubations, sediment was taken from three locations within a stream (i.e., upstream
[Test Case 1a], midstream [Test Case 1b], and downstream [Test Case 1c]) in the Yakima River Basin in Washington,
USA for subsequent laboratory respiration experiments. FTICR-MS was used to determine the OM chemistry from
each sediment sample, resulting in variable formulae being identified in each sample. Formula assignments for all the
samples included herein were completed using formultitude (Tolic et al., 2017).  Total dissolved organic carbon
concentration paired with the FTICR-MS sample and biomass measurements taken at the start of each experiment
were used as the initial concentrations for each of the simulations. Due to the absence of quantitative data related to
how the total carbon mass is distributed between various the OM compounds, the total carbon concentration (on a per-
C basis) was assumed to be split equally between each of the λ bins. The total organic carbon concentration was
distributed into each λ bin using Eq. (13). While this assumption results in equal distribution of carbon between the
bins, consequently, it assigns different initial species concentrations due to varying carbon concentrations between the
molecules.
$[C_{\lambda\text{bin}}]_0 = \dfrac{[TOC]}{n_{\lambda bin} nC_{\lambda bin}}$                               (13)
Where: $[C_{\lambda\text{bin}}]_0$ is the initial species concentration in each λ bin [mol·L$^{-1}$]; $TOC$ is the total organic carbon measured
[mol-carbon·L$^{-1}$]; $n_{\lambda bin}$ is the number of $\lambda$ bins [-]; and $nC_{\lambda bin}$ is the number of carbon molecules in the assumed
formula for the $\lambda$ bins [mol-carbon · mol-molecule$^{-1}$].
Using the Lambda-PFLOTRAN workflow, the FTICR-MS data from each laboratory experiment was digested into
the corresponding λ bins to create the individual reaction network. The Jupyter Notebook for this example is
"Test_Case1-WHONDRS.ipynb" and is available at https://doi.org/10.15485/2281403.

$\mu_{max}$ was fit to the provided experimental oxygen data. The final lambda binned fit, along with corresponding carbon
consumption (individual and total) and aqueous chemistry is displayed in Figure 3 (and in the supporting information
Fig. S1 and S2 for Test Cases 1b [midstream] and 1c [downstream], respectively). To evaluate the use of lambda
binned OM obtained from FTICR-MS (Figure 3), the workflow was also run for a baseline case where $\mu_{max}$ was fit
again, but this time assuming a generic bulk OM form of $CH_2O$ for comparison. The reaction network developed for
a generic OM molecule of $CH_2O$ is shown in Eq. 14.
$2.03\ CH_2O + 0.98\ O_2 + 0.2\ NH_4^+ \rightarrow 1.03\ HCO_3^- + 1.23\ H^+ + 0.4\ H_2O + CH_{1.8}O_{0.5}N_{0.2}$        (14)
This reaction network is used in the Lambda-PFLOTRAN workflow for bulk OM simulations.
Fitted $\mu_{max}$ values for the lambda binned model is 0.25 min$^{-1}$ ($R^2$ = 0.99) and fitted $\mu_{max}$ to the bulk OM $CH_2O$ model
is 0.032 min$^{-1}$ ($R^2$ = 0.96). $V_h$ and CC are fixed at assumed values of 10 m$^3$ and 1 M, respectively in both simulations.

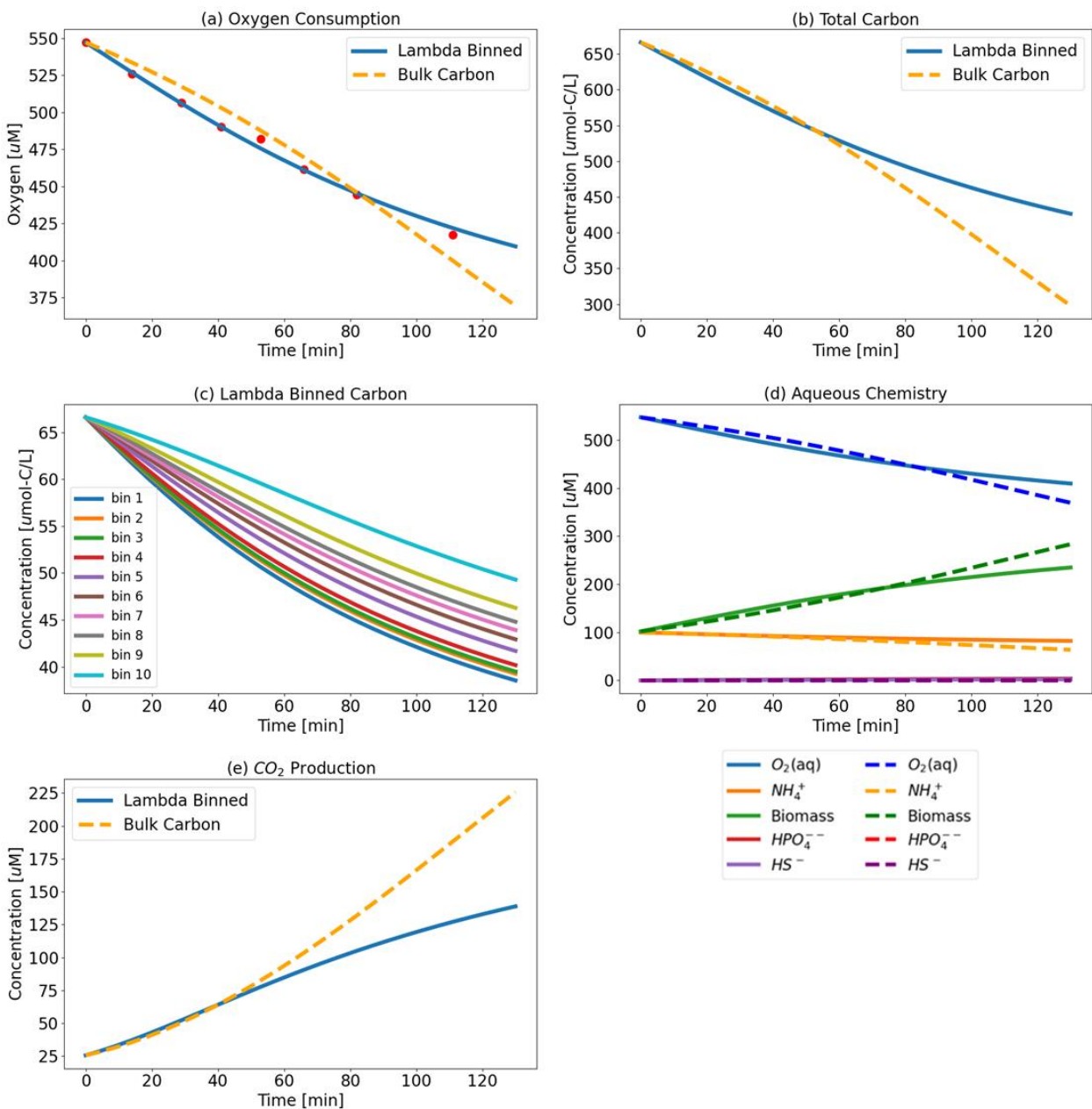

**Figure 3:** Test Case 1a Results – (a) Oxygen Consumption where Lambda-PFLOTRAN workflow was used to fit (blue line) to experimental respiration data (red dots) and (b) Total Carbon Consumption; (c) Individual Organic Matter Consumption by λ bin; and (d) biogeochemistry including $O_2$ (aq) (blue); Biomass (green); $NH_4^+$ (orange); $HS^-$ (purple); and $HPO_4^-$ (red); and (e) $CO_2$ production for the upstream incubation. The dashed orange lines (in a, b and e) show simulation results assuming a generic OM species of $CH_2O$ for comparison.

However, even over the short time frame of this simulation (i.e., only 120 minutes), the difference between assuming the generic $CH_2O$ and using the more detailed organic matter chemistry resulted in different predictions of total carbon and $CO_2$ generation. The bulk OM model predicts more carbon consumption and greater $CO_2$ production than the lambda binned model. The bulk OM model estimates that 50% of the initial total carbon is consumed over the first

120 mins, whereas the lambda binned model predicts 34% consumption. Similarly, the bulk OM model estimates
approximately 35% more $CO_2$ generation as compared the lambda binned model. The effects on aqueous chemistry
over this short duration are more muted, albeit still present.
**3.2 Test Case 2 - Respiration Incubation Experiments.**
Test Case 2 uses soil respiration incubation data from Ward et al. (2023) aimed at investigating the influence of soil
type, oxygen condition (aerobic vs. anaerobic), and seawater exposure (fresh vs. saline) on respiration extent and rate.
For these experiments, temporal measurements were collected for $CO_2$ generation, dissolved organic carbon (DOC),
organic matter formulas via FTICR-MS and other bulk aqueous chemistry (i.e., pH, $NH_4^+$, and other metals and ions)
creating a rich dataset for calibration of system specific lambda model parameters. These incubations were setup by
adding dry soil to the reactor and then adding water (resulting in a soil:water ratio ranging from 1:11 to 1:16). The soil
and water were shaken vigorously for five minutes, and then sampled for the initial time point prior to officially
starting the incubation. For the aerobic experiments, the reactor headspace was cycled every 24 hours to measure $CO_2$
generated but also to ensure the system was kept aerobic; this was only performed five days per week, with no
measurements taken on the weekend due to logistical constraints. Upon experiment completion, the increase in DOC
concentrations indicated organic carbon was being kinetically released from the soil into the aqueous phase over the
course of the 21-day experiment. Similarly, measured $NH_4^+$ concentrations also increased during the experiment. To
address this within our reactive transport model, a source of nitrogen was assumed to be released from the soil as well
($N_{release}$). Both carbon and nitrogen release are included in this example and are assumed to follow a zero-order constant
release rate. Any organic carbon released from the soil was fractionated into each $\lambda$ bin on the same per-carbon basis
assumed for the initial total organic carbon. This was implemented through a dependent function that calculated the
release of carbon into each $\lambda$ bin based on a fitted single bulk $k_{release}$ rate. Mathematically in PFLOTRAN the constant
oxygen conditions were implemented through a gas-liquid partitioning expression with a fast exchange term. These
three additional processes were added to describe the experimental conditions of Test Case 2 more accurately (i.e.,
release of carbon, nitrogen and sustained aerobic conditions); however, a PFLOTRAN input deck can be expanded
and customized to include a host of additional processes and full geochemistry for a specific system of interest. For
instance, aqueous complexation, mineral dissolution and precipitation, sorption, and redox reactions can be added, all
of which can influence the resultant pH and carbon, nitrogen, and other nutrient dynamics.
The workflow was used to fit $\mu_{max}$, $V_h$, CC, $k_{deg}$, as well as $k_{release}$, to the temporal $CO_2$ generation for a single aerobic
soil incubation (Figure 4). The Jupyter Notebook for this example is "Test_Case2-Colloids.ipynb".

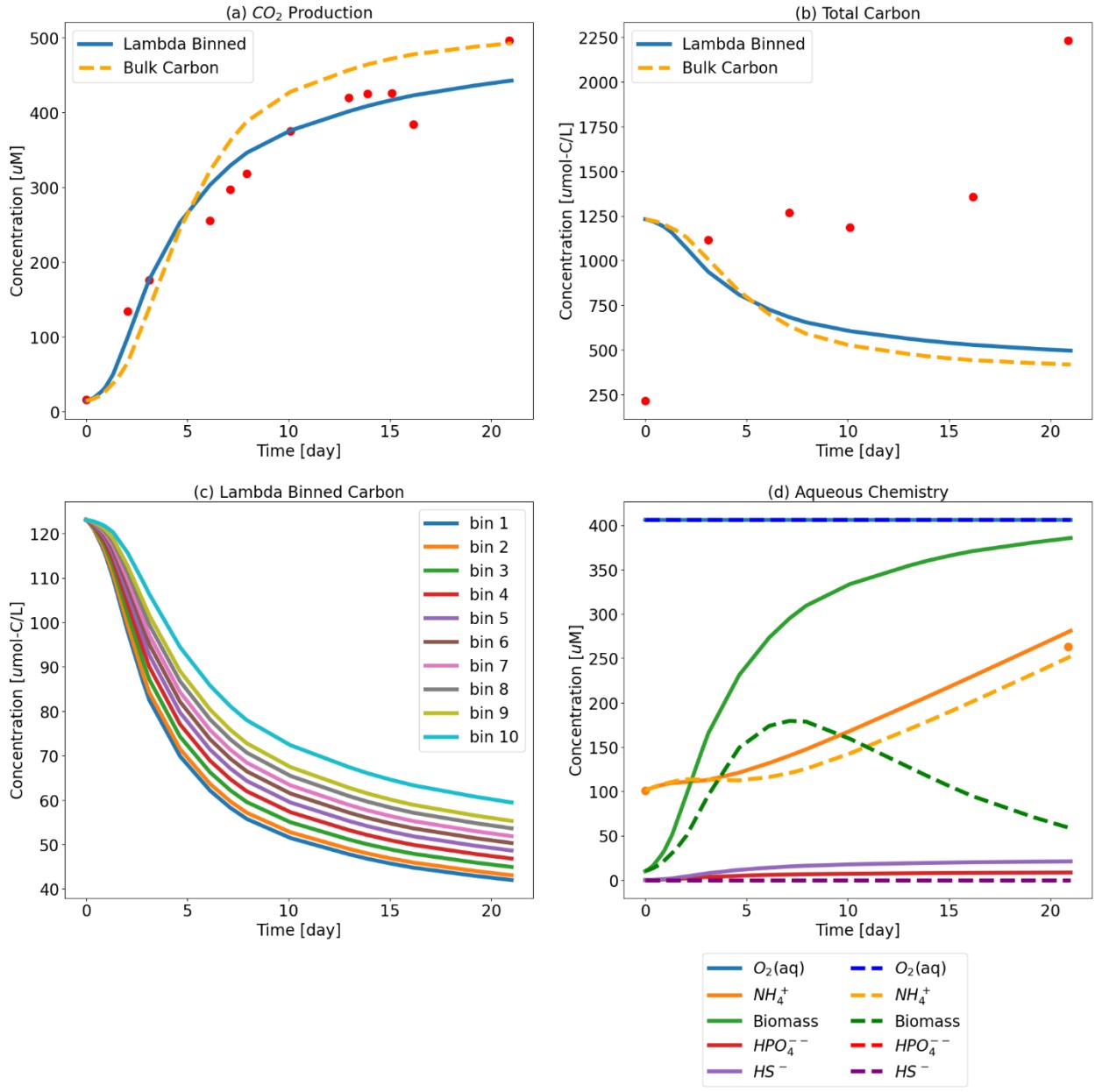


**Figure 4.** Test Case 2 Results – (a) $CO_2$ production where Lambda-PFLOTRAN workflow was used to fit (blue line) to experimental respiration data (red dots) and (b) the corresponding Total Organic Carbon; (c) Individual Organic Matter Consumption by $\lambda$ bin, and (d) the corresponding biogeochemistry including $O_2$ (aq) (blue); Biomass (green); $NH_4^+$ (orange); $HS^-$ (purple); and $HPO_4^{--}$ (red). Dots indicate experimental data. The dashed orange lines in the top two figures show simulation results assuming a generic OM species of $CH_2O$ for comparison. Fitted parameters for lambda binned model are $k_{release} = 5.5 \times 10^{-12}$ day$^{-1}$; $\mu_{max} = 37.6$ day$^{-1}$, $V_h = 5.0$ m$^3$, CC = 0.12 M, and $k_{deg} = 1 \times 10^{-3}$ day$^{-1}$ ($R^2 = 0.953$) and fitted bulk OM $CH_2O$ model values are $k_{release} = 2.0 \times 10^{-12}$ day$^{-1}$; $\mu_{max} = 47$ day$^{-1}$, $V_h = 1.0$ m$^3$, CC = 0.77 M, and $k_{deg} = 0.15$ day$^{-1}$ ($R^2 = 0.909$).


For the purposes for showcasing the workflow, five parameters were estimated in this test case example, and as a
result the models are over parametrized given the amount of data available. Parameter sensitivity over the course of
simulation time is shown in Figure 5 and suggests this system is highly sensitive to $V_h$. It should be noted that both
these model fits are also highly sensitive to the allowable parameter space as user defined by the lower and upper
parameter bounds. In general, parameterization efforts are inherently challenging. For Lambda-PFLOTRAN, which
models microbially mediated processes, it is recommended to initially focus on constraining biomass parameters (i.e.,
CC, $k_{deg}$, and $V_h$) by measuring temporal changes in biomass concentrations. Further, $V_h$ and $\mu_{max}$ are typically highly
sensitive and often correlated. However, since $V_h$ represents the theoretical volume accessible to microbes and cannot
be directly measured, it is suggested to fix $V_h$ within a range of 1-10 m³. If these microbial parameters can be
adequately constrained, focus can shift to $\mu_{max}$, the maximum microbial growth rate, which significantly influences
overall respiration and is expected to exhibit the highest variability across different locations and conditions.

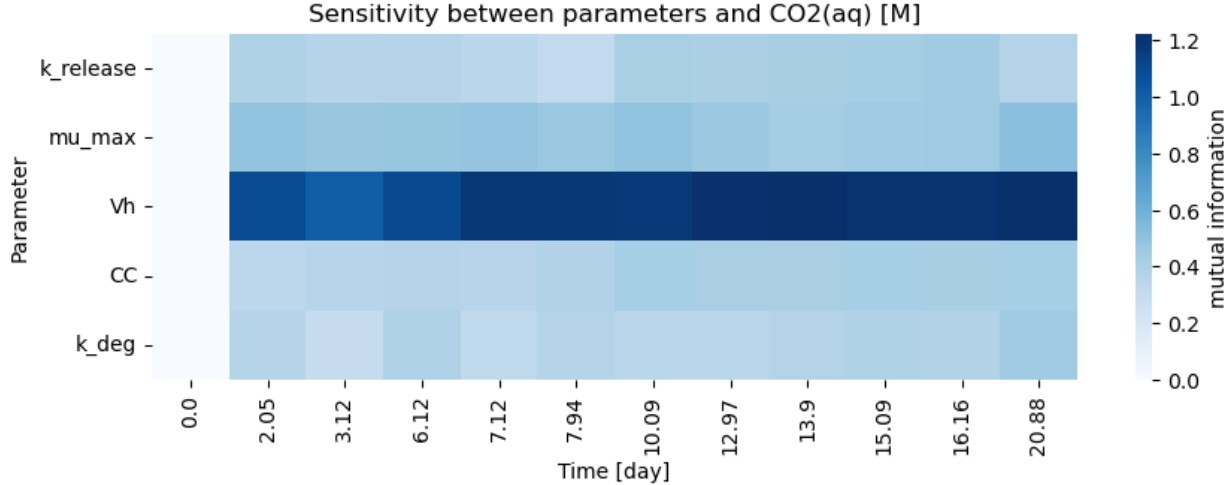


**Figure 5.** Test Case 2 - Sensitivity Analysis Output during Parameter Estimation. The sensitivity of five fitted parameters ($k_{release}$,
$\mu_{max}$, $V_h$, CC, and $k_{deg}$) on temporal aqueous $CO_2$ concentrations as a function of time.
Any additional experimental data, either collected during incubations or through independent experiments (e.g.,
carbon release from the soil in an abiotic system), would be expected to help constraint the model and improve
parameterization. Additionally, it is unclear why the model is unable to capture the total organic carbon behavior in
Test Case 2. One potential explanation is that some of the released organic carbon may not be fully bioavailable and
thus the model may be compensating for this by artificially reducing the concentration of OM available for respiration.
**4 Variability and Impact of Organic Matter Speciation**
The variability in OM speciation was briefly assessed by comparing FTICR-MS data from Test Cases 1 and 2. Each
identified OM species was classified into one of nine compound classes. For Test Case 1, the average of the three Test
Case 1 samples (1a - upstream, 1b - midstream, and 1c - downstream) was computed. The predominant classes were
proteins ($34 \pm 1\%$), lignin ($26 \pm 1\%$), and lipids ($13 \pm 2\%$), with the errors representing the standard deviation among
the Test Case 1a-c samples. The low standard deviation suggests consistent reproducibility in OM speciation for
samples taken from nearby locations. In contrast, OM in Test Case 2 was primarily composed of lignin (37.4%) and
concentrated hydrocarbons (32%). The full distribution of compound classes is presented in Figure 6.

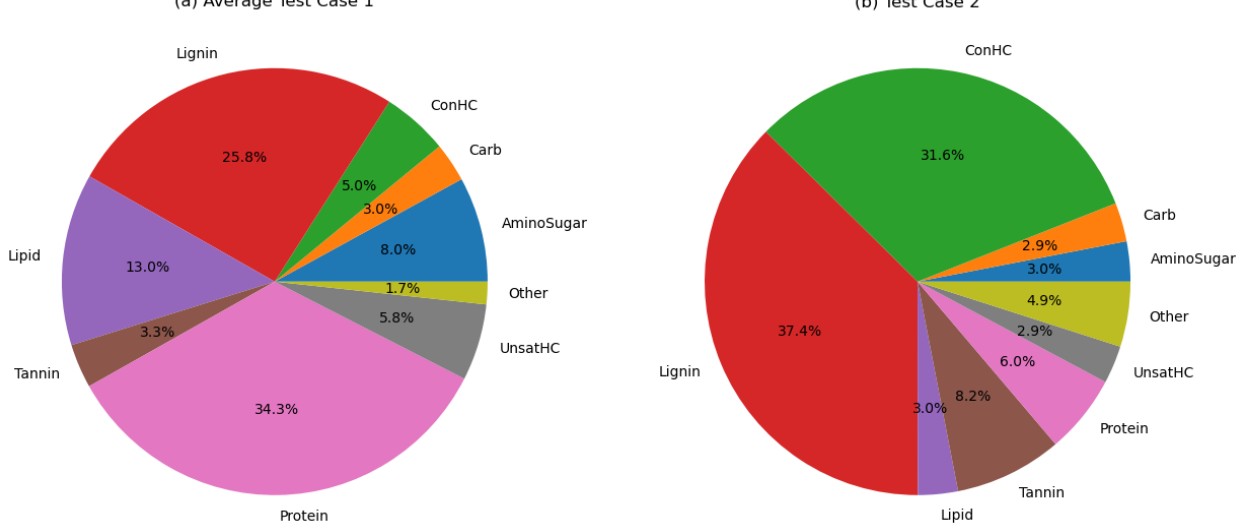


**Figure 6.** Distribution of Organic Matter Compound Classes: (a) Test Case 1 and (b) Test Case 2.
Note: Test Case 1 is the average of Test Case samples 1a-c. ConHC = Condensed Hydrocarbon; UnsatHC = Unsaturated
Hydrocarbon

The influence of the sample OM speciation on the λ binned reaction networks was also assessed. Figure 7 illustrates
the impact of OM speciation on the corresponding λ binned reaction networks, with three key observations. First, the
variability in OM speciation between different samples is evident when comparing Test Case 1 and Test Case 2. To
enhance visual clarity, the range of Test Case 1 samples (1a-c) is depicted as a grey shaded region, showing the spread
between the minimum and maximum values of the three samples. For Test Case 2, data from the single FTICR-MS
sample is represented by blue dots. Test Case 1 and 2 have distinct λ derived reaction networks as indicated by the
little overlap between the grey region and the blue dots in Figures 7b-i.

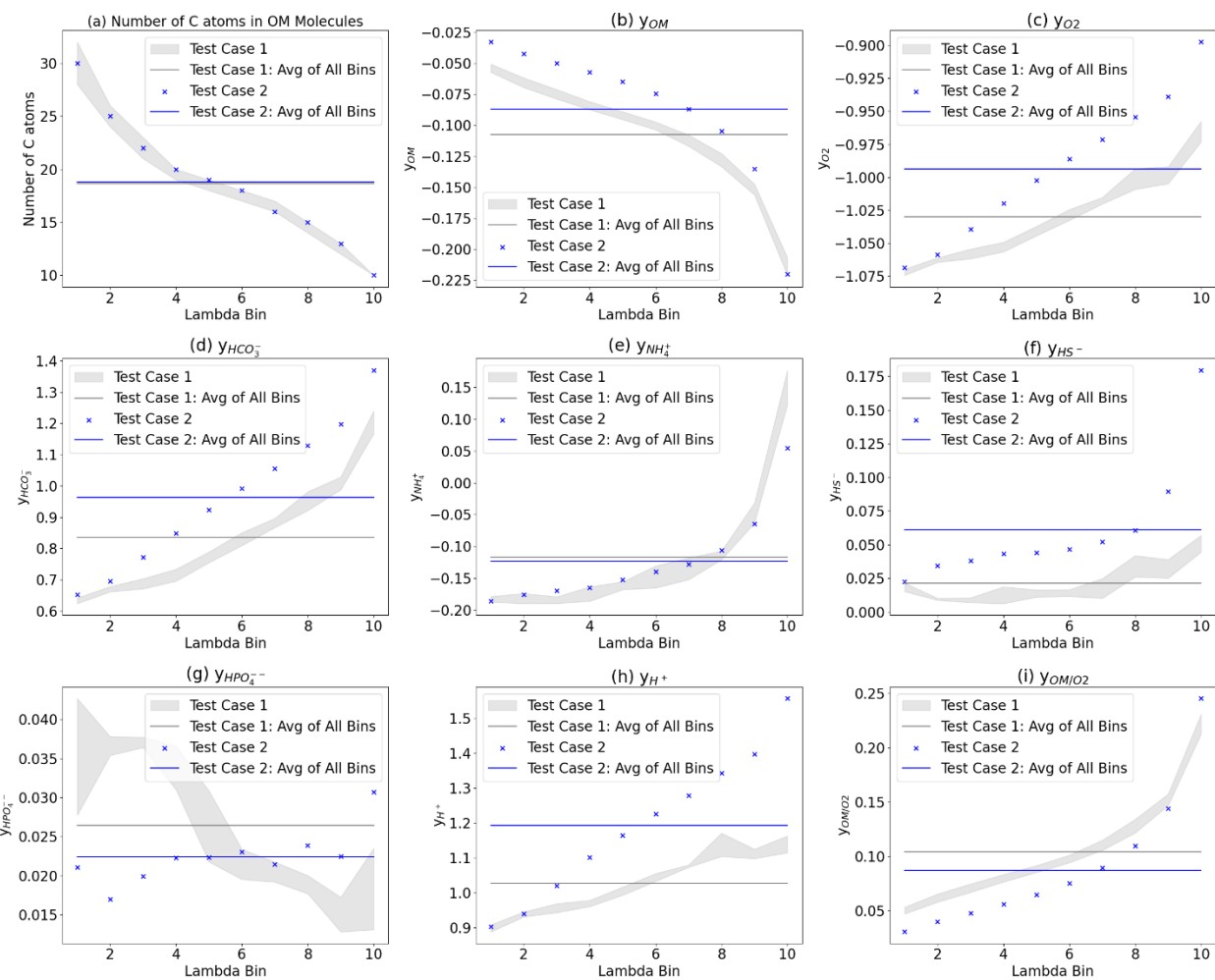

**Figure 7.** Comparison of Lambda-Binned Reaction Network Parameters: (a) number of carbons in the OM; stoichiometric coefficient, $y$, for (b) OM, (c) $O_2$, (d) $HCO_3^-$, (e) $NH_4^+$, (f) $HS^-$, (g) $HPO_4^{--}$, (h) $H^+$; and (i) ratio of OM/$O_2$ coefficients for Test Case 1a-c (grey dots); the average of all $\lambda$ bins for Test Case 1 (grey line); Test Case 2 (blue x); and the average of all $\lambda$ bins for Test Case 2 (blue line). The grey shaded area highlights the range of values for Test Case 1a-c for better visual comparison.

Second, the $\lambda$ binning process captures the OM speciation variation within a sample. To illustrate this intrasample variability, a line representing the average of all $\lambda$ bins is shown on Figure 7 (grey line for Test Case 1, blue line for Test Case 2). The difference between the reaction network coefficients (vertical axis) for the $\lambda$ binning (grey shaded area and blue dots) and the Test Case average lines highlights the extent of this variability. Finally, although the $\lambda$ binning process resulted in a similar number of carbon atoms to OM molecules within each $\lambda$ bin for both test cases (Figure 7a), the resulting stoichiometric coefficients in the reaction networks differ significantly (Figures 7b-h). These stoichiometric differences lead to variations in biogeochemical outcomes, such as OM-to-oxygen utilization ratios during aerobic respiration (Figure 7i). These differences are due to the additional elements beyond carbon in the OM molecules (i.e., nitrogen, oxygen, sulfur, hydrogen, and phosphorus).

To further assess and isolate the effect of OM speciation, extended forward simulations were performed by only
varying FTICR-MS input data (Figure 8). FTICR-MS samples from Test Cases 1a-c and Test Case 2 were tested.
These simulations replicate Figure 3 (i.e., Test Case 1a conditions and fitted $\mu_{max}$ values) with the expectation of OM
speciation, and demonstrate the significant impact of OM chemistry and speciation on overall predicted behavior,
especially over longer time periods.

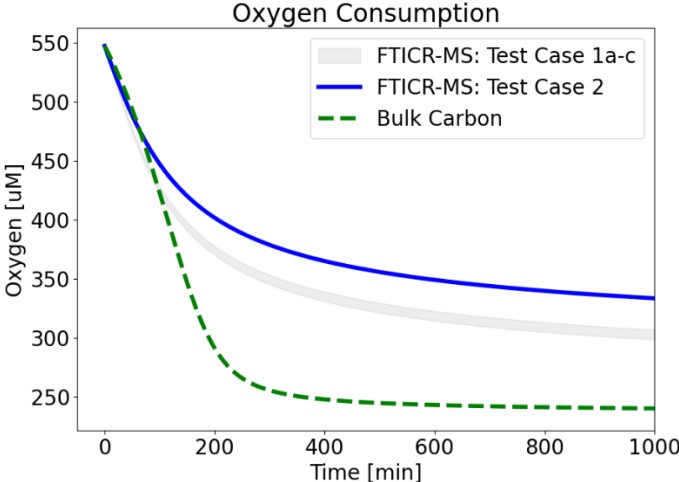


**Figure 8.** Influence of OM Speciation on Oxygen Consumption. FTICR-MS data from Test Cases 1a-c (grey shaded area), and
Test Case 2 (blue line) were used as inputs. Bulk $CH_2O$ OM (green line) was also plotted for reference. Best fit $\mu_{max}$ values to Test
Case 1a were used (i.e., lambda binned $\mu_{max} = 0.25$ min$^{-1}$; bulk OM $\mu_{max} = 0.032$ min$^{-1}$).

The clear variability in OM speciation, differences between a generic OM reaction network and one informed by
FTICR-MS, and the impact of OM chemistry on biogeochemical predictive simulations underscore the importance of
incorporating site-specific OM chemistry informed by ultra high resolution characterization into biogeochemical
models.
**5 Conclusions**
Overall, Lambda-PFLOTRAN workflow provides an important linkage between molecular scale organic matter
characterization and reactive transport simulations. This workflow allows for the influence of organic matter
composition to be utilized within simulators to provide a more comprehensive understanding of the system chemistry
and behavior, moving beyond the standard assumption of bulk organic matter chemistry and composition. While there
are current limitations due to how composition is characterized and quantified, this workflow connecting
characterization information to simulations is an important advancement that can be refined as these laboratory
techniques improve over time.
One of the major limitations surrounding this method, is the lack of understanding of organic matter compound
bioavailability, resulting in a large conceptual gap as to how various organic carbon compounds may be utilized by
microbes. In the absence of such information, all identified organic matter molecules are assumed to have equal
bioavailability within this modeling framework when, in reality, compounds will exhibit varying degrees of
bioavailability depending on factors such as associated size fraction, carbon pool, and environmental factors (Schmidt
et al., 2011; Ahamed et al., 2023). Until improved understanding is established to discern individual compound
bioavailability, this will remain as a limitation.
Another limitation of this method resides around the analytical limitations of organic carbon characterization and
quantification. For instance, FTICR-MS focuses on water soluble organic matter which may provide a basis in the
types of carbon identified by this technique (Tfaily et al., 2017). Additionally, as mentioned previously, FTICR-MS
is qualitative, it does not provide structural information and will not differentiate between different isomers that have
the same molecular formulas, it is only able to identify molecular formula is present or absent and not the concentration
associated with each peak. Here, this has been addressed by assuming equal distribution of total carbon between the
formulas within each $\lambda$ bin on a per-carbon basis. This caveat can be easily updated in the workflow if new analytical
advances are made that provide more quantitative information. Some existing approaches could be suitable for this
type of modeling such as using quantitative biomarkers that cover major compound classes (Kim and Blair, 2023);
but further advances in obtaining both high resolution and quantitative OM characterization would greatly aid in how
we understand and model ecosystems.

**Acknowledgements:**

This research was performed under a variety of interdisciplinary projects including the U.S. Department of Energy (DOE) sponsored Office of Science, Office of Biological and Environmental Research (BER), Environmental System Science (ESS) Program, IDEAS-Watersheds, River Corridor Scientific Focus Area (SFA), the Environmental Molecular Sciences Laboratory User Facility sponsored by the Biological and Environmental Research program under Contract No. DE-AC05-76RL01830, and COMPASS-FME, a multi-institutional project supported by DOE-BER as part of the Environmental System Science Program. This study used data from the Worldwide Hydrobiogeochemistry Observation Network for Dynamic River Systems (WHONDRS). This paper describes objective technical results and analysis. The work was performed at the Pacific Northwest National Laboratory (PNNL). PNNL is operated for DOE by Battelle Memorial Institute under contract DE-AC05-76RL01830. This paper describes objective technical results and analysis. Any subjective views or opinions that might be expressed in the paper do not necessarily represent the views of the U.S. Department of Energy or the United States Government.

**Code Availability:**

The source code, installation requirements, example test case notebooks, and associated data are available in ESS DIVE at https://doi.org/10.15485/2281403

**Author Contribution:**

KM: conceptualization, formal analysis, methodology, software, writing- original draft preparation; PJ: methodology, software, writing- original draft preparation; GH: methodology, software, writing-review & editing; TA: data curation, software, writing-review & editing; HS: methodology, writing-review & editing; RK: supervision; NW: supervision, writing-review & editing; MB: investigation; RC: investigation; QZ: investigation; VG: investigation, data curation; AR: investigation; XC: conceptualization, investigation, writing-review & editing

**Competing Interests:** The authors declare that they have no conflict of interest.

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
