# Peer review of "Lambda-PFLOTRAN 1.0: Workflow for Incorporating Organic Matter Chemistry Informed by Ultra High Resolution Mass Spectrometry into Biogeochemical Modeling"

_Geoscientific Model Development, 2024_

## Author Response (AR1)

**GMD-2024-34 Response to Reviewer Comments**

Lambda-PFLOTRAN 1.0: Workflow for Incorporating Organic Matter Chemistry Informed by Ultra High Resolution Mass Spectrometry into Biogeochemical Modeling

We thank the reviewers for their thoughtful review and comments. In response, we have made substantial improvements to the manuscript. Most notably, we have added a new section exploring the influence and variability of OM chemistry (**Section 4- Variability and Impact of Organic Matter Speciation**), which we believe has significantly strengthened our manuscript. Additional responses to specific reviewer comments are detailed below.

**Reviewer 1:**

**R1.1** The presented work is a good contribution to improving the reactive transport modeling approach. It develops a representative reaction network based on substrate-explicit thermodynamic modeling and completes a biogeochemical simulation with a reactive transport model.

However, I think the current version cannot be considered for publication in GMD. There are some possible reasons:

**R1.2** Poor model validation. A strict validation checks whether a new model approach reflects the actual or desired physical and biochemical behavior, considering using a reactive transport model. In this study, I noticed that there are very limited observations. For example, the work only involved limited observed $O_2$ (Fig.4a), $CO_2$ (Fig.5a), and total organic carbon datasets (Fig.5b). I also noticed that the model cannot capture observed total organic carbon (Fig.5b). Incomplete simulation experiments. The work just considered two test cases (oxygen depletion and respiration). Too small a test size may lack sufficient statistical power to detect new model capabilities. Poor analysis. The work shows limited biogeochemical analysis. An advantage of RTM is its ability to elucidate complex biogeochemical dynamics. For example, How do oxygen dynamics and total carbon influence the variations of pH and C, N, and P species over time (Fig.5a)? Although I understand this work is technical, the justification for these time-series variations should be carefully discussed.

We understand the reviewer's comment on the importance of model validation. We agree that more complete data for the test cases and additional datasets would be greatly beneficial, yet high-resolution organic matter characterization (e.g., FTICR-MS) is not commonly measured, which makes it not feasible to thoroughly validate the model given the lack of required data with both incubation and organic matter speciation. We hope that building this modeling framework will encourage others to measure organic matter speciation via FTICR-MS in the future. As more data becomes available, the Lambda-PFLOTRAN workflow can be further validated and refined.

**We have clarified the scope of this paper on line 80 (tracked changes version),** emphasizing that our work aims to detail a new capability and associated workflow for incorporation of organic matter characterization into reactive transport models. Model validation is important and new data collection could be designed to make it possible in future studies.

The illustrative model results presented here consider only aerobic respiration through the FTICR-MS informed reaction network. If desired, the PFLOTRAN input deck can be expanded and customized to include additional processes and full geochemistry.

We have added the following statement on **lines 328-333 (tracked changes version)** to capture this point:

> **"Three additional processes were added to describe the experimental conditions of Test Case 2 more accurately (i.e., release of carbon, nitrogen and sustained aerobic conditions); however, a PFLOTRAN input deck can be expanded and customized to include a host of additional processes and full geochemistry for a specific system of interest. For instance, aqueous complexation, mineral dissolution and precipitation, sorption, and redox reactions can be added, all of which can influence the resultant pH and carbon, nitrogen, and other nutrient dynamics."**

**Reviewer 2:**

**R2.1** This manuscript describes a new workflow for integrating detailed organic matter data from FTICR-MS into a reactive transport modeling framework for simulating biogeochemical interactions. The workflow includes sensitivity analysis and parameter estimation capabilities. I think this has the potential to be a highly valuable tool for connecting biogeochemical models with organic matter measurements at a level of chemical detail that has historically been a major challenge for biogeochemical model-data integration. The workflow covers many of the pain points for this type of model-data connection, including aggregating large numbers of compounds into a smaller, tractable set of representative compounds, assigning meaningful model parameters to those compounds, and converting the resulting reaction network into a reactive transport model simulation configuration automatically.

The manuscript describes the underlying assumptions and theoretical framework in clear terms and acknowledges important caveats such as the lack of quantitative information about the relative amounts of different compounds. The steps for using the workflow are also clearly described, which fits with the goal of making the workflow accessible and reproducible for other scientists. The two provided test cases are useful concrete examples of the workflow, important parameters, and useful data types for parameter estimation.

**R2.2** I did think that the results shown from the test cases did not fully make the case for how the FTICR-MS data add value to the resulting simulations. While the comparison of the full reaction network to simulations using multiple compounds does show a difference in overall predicted concentrations relative to a simplified reaction network with only one organic matter compound, it is difficult to see how the high resolution of chemical compounds, which is the hallmark of the FTICR-MS analysis, specifically contributes to the simulation results.

We have revised **Figure 3,** the associated figure description, and companion text to help highlight the impact of OM speciation for Test Case 1 **(lines 278-302- tracked changes version).**

**R2.3** It would be helpful to include a visualization of the distribution of compounds, how they translate to reaction networks with different properties, and how that ultimately affects the model results. For example, is the reaction network derived from the Test Case 1 FTICR-MS measurements meaningfully different from that of Test Case 2 in terms of its stoichiometry or distribution of reaction rates? Does the specific FTICR-MS data from each experiment yield better model results than FTICR-MS data from a different experiment? What if the null hypothesis was 10 compounds derived from a different set of measurements, rather than 1 generic compound? This comparison could more directly show the value of those measurements from a particular experiment.

We have added a new section titled **"Variability and Impact of Organic Matter Speciation"** to address these comments. Section 4 explores the differences in OM chemistry between samples (Figure 5) and the effects of OM chemistry on the lambda-derived reaction networks (Figure 6). Additionally, this section also examines the influence of OM chemistry through forward simulations, where only FTICR-MS input data is varied (Figure 7).

**R2.6** Line 97: is $y_{OM}$ here the same as $y_{OC}$ in Equation 2?

Thank you for catching this. We have updated so it reads $y_{OM}$.

**R2.7** Line 101: I think "1000" is missing from this sentence

Line 101 has been updated.

**R2.8** Line 123: HS should be $HS^-$

Thank you for the careful read. Updated on line 123.

**R2.9** Line 192: Given the importance of specifying parameters to be estimated for the workflow, I think it would be helpful to list all the relevant parameters in one table, with explanations of what they mean and perhaps with suggestions for what kind of measurements could constrain them or what would be a good context for including them in the parameter estimation and what reasonable ranges of values might be.

Model parameters are defined in the Methods section (Section 2). While we understand the interest in gaining further insight into optimal model parameterization, we believe this is beyond the scope of the current manuscript. However, we agree that additional work focused on parameterization, experimental design, and data collection would be valuable and could be pursued in future research.

**R2.10** Section 2.3.4: When discussing the parameter optimization, it would be helpful to start with some explanation of what kind of observations are useful for comparing with the model. While examples are provided in the test cases, I think some general explanation earlier on would be helpful as well.

We understand the interest in extending this manuscript to include parameterization and optimization; however, we believe this goes beyond the scope of a model description paper.

**R2.11** Line 256: I'm curious how accurate the assumption of equal division of C mass across bins is. Do any data exist that could inform this? The approach might be quite sensitive to this assumption.

We agree with the reviewer about the potential impact of the equal division of carbon mass across bins. Unfortunately, to our knowledge, there is currently no information available regarding the assumption of mass distribution among organic matter species. This presents an excellent opportunity for follow-up research. Any data-backed updates on mass distribution can be easily incorporated into the Lambda-PFLOTRAN framework.

**R2.12** Figure 3: I was confused why k_deg and C_inhibit are shown in this figure, even though they aren't part of the parameters being tested. And I did not understand the significance of the time dimension. Does this indicate whether model ensemble members are diverging or converging over the course of the simulation? It could use some more explanation of how to interpret that part.

In response, the lambda-PFLOTRAN sensitivity output is now being shown for Test Case 2 in Figure 5. Additional details have been added on lines 350 -353 (tracked changes version) as well.

**R2.13** Figure 5: The model generally fails to reproduce the observed total carbon concentrations. I might guess that the assumption of how much carbon gets solubilized from the solid state over time, or the characteristics of that soluble carbon, are not correct for this system. I would not necessarily expect a model to get every aspect of the system right, especially if it relates to a steady state process involving solid organic matter, which this framework is not designed to replicate. But I think this should be at least acknowledged and explained in the text and currently it feels like this part of the result is ignored.

We have added our hypothesis for why the model is unsuccessful at capturing the total organic carbon concentrations on lines 359 – 361 (tracked changes version).

---

## Author Response (AR2)

We thank for the editor for the comments and edits. We have addressed the specific comments as follows:

R1) Both reviewers asked for suggestions or guidelines to address the parameterization step. Although I agree with you that a detailed parameterization description is beyond the scope of the article, the request is much simpler than this. Simply provide, based on your experience, a set of recommendations for users about how to address the parameterization problem. For example, what parameters should receive priority, what parameters tend to be correlated among each other, and which ones seem to be less sensitive. This is just to help new users get started with this difficult task.

We have added the following text on lines 332-338 as initial guidance related to parameterization.

"In general, parameterization efforts are inherently challenging. For Lambda-PFLOTRAN, which models microbially mediated processes, it is recommended to initially focus on constraining biomass parameters (i.e., CC, $k_{deg}$, and $V_h$) by measuring temporal changes in biomass concentrations. Further, $V_h$ and $\mu_{max}$ are typically highly sensitive and often correlated. However, since $V_h$ represents the theoretical volume accessible to microbes and cannot be directly measured, it is suggested to fix $V_h$ within a range of 1-10 m³. If these microbial parameters can be adequately constrained, focus can shift to $\mu_{max}$, the maximum microbial growth rate, which significantly influences overall respiration and is expected to exhibit the highest variability across different locations and conditions."

R2) Line 170 (tracked changes version). bias instead of basis?

We have made this update on line 409 (tracked changes version).

R3) Test case 1 presents a comparison with a 'bulk SOM model' or 'bulk carbon' model, but I don't recall seeing a description of this model. Do you have a description of this model in the previous sections?

We have moved the description of the generic bulk carbon reaction network up to line 274 – 277 (tracked changes version).

R4) Figure 6. Replace 'carbons' for 'C atoms'.

Updated Figure 6 accordingly.

R5) When clicking on the link https://doi.org/10.15485/2281403 I get a blank page. Can you check whether this is working properly?

The link is working properly for us. Please let us know if this is still an issue though.

R6) One comment on your method, which I don't expect you to address in a revised version. The fact that the input data doesn't provide information on how the total carbon mass is distributed

among the different lambda bins seems to be as a major limitation of this approach. However, the approach allows you to compute entropies, and you have some entropy formulas in equation 11. Have you thought about using entropy maximization to obtain a plausible distribution of the lambda bins? This seems to me as a good potential approach to this with this limitation of the method.

Thank you for this idea! It is very interesting and worth pursuing.